# Immunostimulatory Effect of Flagellin on MDR-*Klebsiella*-Infected Human Airway Epithelial Cells

**DOI:** 10.3390/ijms25010309

**Published:** 2023-12-25

**Authors:** Christine C. A. van Linge, Katina D. Hulme, Hessel Peters-Sengers, Jean-Claude Sirard, Wil H. F. Goessens, Menno D. de Jong, Colin A. Russell, Alex F. de Vos, Tom van der Poll

**Affiliations:** 1Center for Experimental and Molecular Medicine, Amsterdam University Medical Centers, University of Amsterdam, 1012 WP Amsterdam, The Netherlandsa.f.devos@amsterdamumc.nl (A.F.d.V.); t.vanderpoll@amsterdamumc.nl (T.v.d.P.); 2Amsterdam Infection & Immunity Institute, 1105 AZ Amsterdam, The Netherlands; 3Department of Medical Microbiology & Infection Prevention, Amsterdam University Medical Centers, University of Amsterdam, 1012 WP Amsterdam, The Netherlands; 4Center for Infection and Immunity of Lille, Institut Pasteur de Lille, INSERM U1019, CNRS UMR9017, CHU Lille, University Lille, 59000 Lille, France; 5Department of Medical Microbiology and Infectious Diseases, Erasmus University Medical Center, 3015 GD Rotterdam, The Netherlands; 6Department of Global Health, School of Public Health, Boston University, Boston, MA 02215, USA; 7Division of Infectious Diseases, Amsterdam University Medical Centers, University of Amsterdam, 1012 WP Amsterdam, The Netherlands

**Keywords:** flagellin, human bronchial epithelial cells, *Klebsiella pneumoniae*, multi-drug resistance, pneumonia, antibiotics, meropenem

## Abstract

Pneumonia caused by multi-drug-resistant *Klebsiella pneumoniae* (MDR-*Kpneu*) poses a major public health threat, especially to immunocompromised or hospitalized patients. This study aimed to determine the immunostimulatory effect of the Toll-like receptor 5 ligand flagellin on primary human lung epithelial cells during infection with MDR-*Kpneu*. Human bronchial epithelial (HBE) cells, grown on an air–liquid interface, were inoculated with MDR-*Kpneu* on the apical side and treated during ongoing infection with antibiotics (meropenem) and/or flagellin on the basolateral and apical side, respectively; the antimicrobial and inflammatory effects of flagellin were determined in the presence or absence of meropenem. In the absence of meropenem, flagellin treatment of MDR-*Kpneu*-infected HBE cells increased the expression of antibacterial defense genes and the secretion of chemokines; moreover, supernatants of flagellin-exposed HBE cells activated blood neutrophils and monocytes. However, in the presence of meropenem, flagellin did not augment these responses compared to meropenem alone. Flagellin did not impact the outgrowth of MDR-*Kpneu*. Flagellin enhances antimicrobial gene expression and chemokine release by the MDR-*Kpneu*-infected primary human bronchial epithelium, which is associated with the release of mediators that activate neutrophils and monocytes. Topical flagellin therapy may have potential to boost immune responses in the lung during pneumonia.

## 1. Introduction

Pneumonia, an acute respiratory infection affecting the bronchial tree and alveoli of the lungs, is a major health problem, associated with a high morbidity and mortality worldwide [1]. Pneumonia is classified as community-acquired pneumonia (CAP), with *Streptococcus* (*S.*) *pneumoniae* as the most common causative pathogen, or hospital-acquired pneumonia (HAP) [2], with *Staphylococcus aureus*, *Pseudomonas* (*P.*) *aeruginosa*, or *Enterobacteriaceae*, such as *Klebsiella* (*K.*) *pneumoniae*, as frequently found microorganisms [2]. In the last decade, *K. pneumoniae* has also been recognized as an emerging cause of CAP in certain areas of the world [3]. Antimicrobial resistance poses a major public health threat with an estimated 1.27 million attributable deaths in 2019 [4]. *K. pneumoniae* was found to be one of the six leading pathogens contributing to these deaths, with carbapenem-resistant *K. pneumoniae* responsible for 50,000–100,000 attributable deaths [4].

The epithelium of the respiratory tract forms the first line of defense against invading pathogens. Besides acting as a physical barrier, it is well established that the airway epithelial cell layer plays a central role in host defense through mucociliary clearance, secretion of antimicrobial agents, and the release of chemoattractants that can recruit immune cells to the site of infection [5,6]. The bronchial epithelium comprises, amongst others, ciliated cells, mucus-secreting cells, and basal cells, which act as resident stem cells [5]. Recently, it was found that bronchial basal cells, isolated from human lung tissue, can be expanded and differentiated in vitro into pseudo-stratified human bronchial epithelial (HBE) cell layers when grown on a permeable filter with air on the apical side and cell culture media on the basolateral side, a so-called air–liquid interface (ALI), mimicking the airways. The validity of this culture method is supported by studies on the structure, gene expression, and function of the epithelium, including secretion of mucus and inflammatory mediators, and has been widely employed to complement in vivo studies [7,8].

Respiratory epithelial cells express multiple pattern recognition receptors (PRRs), such as Toll-like receptors (including TLR1-10), and NOD-like receptors, which can sense microbe-associated molecular patterns [5,7]. TLR5 is a key PRR on the airway epithelium, for the recognition of flagellin [9]. Flagellin is the structural component of the flagellum, a surface ligament dedicated to bacterial motility. Bacteria such as *P. aeruginosa* or *Salmonella* species are equipped with a flagellum to propel through mucus layers in the gut or lung [10]. Flagellin-mediated activation of the TLR5 signaling cascade results in the secretion of inflammatory mediators and recruitment of phagocytes such as neutrophils [10]. Studies by our group and others showed that stimulation of human airway epithelial cells with flagellin or flagellated bacteria induces secretion of inflammatory mediators in vitro [11,12,13]. Furthermore, in vivo studies with TLR5-deficient mice revealed a critical role for TLR5 on lung epithelial cells in the immune response upon intranasal exposure to flagellin [14,15]

To aid in host defense against antibiotic-resistant bacteria, new drugs and treatment strategies are being developed [2,16]. Recently, it was found in an experimental CAP model in mice that administration of flagellin via the airways in combination with systemic antibiotic treatment reduced the burden of antibiotic-susceptible, as well as antibiotic-resistant, *S. pneumoniae* in the lung and spleen and increased survival [17,18]. These studies showed that flagellin in combination with antibiotics augmented the chemokine response in the lungs as compared to treatment with antibiotics alone, which was associated with increased neutrophil recruitment to the lung [17,18]. To determine whether immune-stimulatory therapy may have potential in pneumonia evoked by antibiotic-resistant bacteria, here, we sought to study the immunomodulatory effect of flagellin in an in vitro model of primary pseudostratified HBE cells infected with carbapenem-resistant (non-flagellated) *K. pneumoniae*. Since antibiotics are the mainstay of therapy for bacterial pneumonia [2], we investigated the effect of flagellin also in combination with meropenem, a broad-spectrum β-lactam antibiotic, often used as empirical therapy [19]. To mimic the clinical situation, we commenced treatment with flagellin and/or meropenem during ongoing infection with MDR-*Kpneu* on the apical side of the HBE cells (representing the airways) and applied meropenem to the basolateral medium (representing the inside of the body) and flagellin on the apical side in view of topical therapy. The results of our study reveal that flagellin treatment of infected HBE cells augments the expression of antimicrobial factors and chemokines and triggers the secretion of inflammatory mediators that activate neutrophils and monocytes.

## 2. Results

### 2.1. Flagellin and Meropenem Induce the Expression of Antimicrobial Proteins in MDR-Kpneu-Infected HBE Cells

To determine whether flagellin augments the innate immune responses of infected HBE cells, we first investigated whether flagellin triggered the production of antimicrobial proteins and reduced the outgrowth of MDR-*Kpneu*. To assess whether flagellin impacts the antibacterial response, HBE cells were infected with viable MDR-*Kpneu* (1000 colony-forming units (CFUs)) or administered an equal volume of phosphate-buffered saline (PBS; control) on the apical surface. After 6 h of incubation, HBE cells were treated with meropenem or medium on the basolateral side and flagellin or PBS on the apical side (Figure 1A). Of note, the MDR-*Kpneu* strain used in the current study is susceptible to meropenem at a concentration of 50 µg/mL (Appendix A). Flagellin enhanced the expression of the genes encoding β-defensin 4a (*DEFB4A*), also known as β-defensin 2, peptidase inhibitor 3 (*PI3*), and calprotectin (*S100A8* and *S100A9*) in HBE cells collected 24 h after initiation of the experiment (infection or control); these flagellin effects were similar in uninfected and MDR-*Kpneu*-infected HBE cells (Figure 1B). Strikingly, while infection with MDR-*Kpneu* alone did not impact the expression of these genes, meropenem treatment of MDR-*Kpneu*-infected HBE cells increased *DEFB4A*, *PI3*, *S100A8*, and *S100A9* expression to a similar level as flagellin alone, and this was not further increased by simultaneous flagellin exposure. Meropenem did not modify the expression of these genes in the absence of MDR-*Kpneu*, excluding an intrinsic immune-enhancing effect of this antimicrobial agent. S100A8 and S100A9 proteins together form a heterodimer named calprotectin, a protein implicated in host defense against bacteria including *Klebsiella* [20,21]. Thus, we next measured calprotectin concentrations in the basolateral supernatant and apical wash of HBE cells.

As compared to the apical wash, calprotectin levels in the basolateral medium were low and not influenced by either flagellin or meropenem (Appendix A). Calprotectin levels in the apical wash, however, were increased in HBE cells exposed to flagellin alone (Figure 1C), but flagellin did not augment calprotectin levels in MDR-*Kpneu*-infected HBE cells, irrespective of meropenem treatment.

After 24 h of infection, bacterial numbers were analyzed to determine the outgrowth of MDR-*Kpneu* (Figure 1D). CFU counts were >10^8^ in all groups, showing a massive outgrowth of MDR-*Kpneu* on the apical side of the HBE cells (Figure 1D). Flagellin or meropenem by itself did not affect bacterial counts at this site, and the combination treatment showed only a trend toward reduced bacterial outgrowth. Strikingly, no bacteria were detected in the basolateral medium. Taken together, these results suggest that exposure to flagellin induces the expression of antimicrobial proteins in MDR-*Kpneu*-infected HBE cells but does not markedly impact the antibacterial effect of these cells against MDR-*Kpneu.*

### 2.2. Flagellin and Meropenem Induce the Secretion of Inflammatory Mediators by MDR-Kpneu-Infected HBE Cells

Previously, we demonstrated that flagellin induces the production and polarized secretion of chemokines by HBE cells cultured on ALI [12]. Here, we determined the additive effect of flagellin on *CXCL1*, *CXCL8*, and *CCL20* expression by HBE cells that were infected with MDR-*Kpneu* and subsequently treated with meropenem (Figure 2A).

Analysis of mRNA levels confirmed that flagellin treatment alone increased the expression of *CXCL1*, *CXCL8*, and *CCL20* in HBE cells as compared to medium-treated cells (Figure 2B).

Exposure of HBE cells to MDR-*Kpneu* alone slightly increased the expression of these chemokines as compared to controls, and the expression of *CXCL1* and *CCL20* was significantly enhanced by meropenem treatment. Flagellin augmented the expression of *CXCL1* and *CCL20* in MDR-*Kpneu*-infected cells but did not further enhance their expression in combination with meropenem. Next, we measured the secretion of CXCL1, CXCL8, and CCL20 protein in the basolateral medium of MDR-*Kpneu*, meropenem, and flagellin-exposed HBE cells (Figure 2C). The secretion pattern of CXCL1, CXCL8, and CCL20 largely mimicked the mRNA expression pattern of these chemokines, with distinct induction of CXCL1 and CCL20 by flagellin alone and minimal release of all three chemokines by MDR-*Kpneu* alone. Furthermore, flagellin augmented the secretion of CXCL1, CXCL8, and CCL20 by MDR-*Kpneu*-infected cells but did not further increase secretion when combined with meropenem treatment. Although we also demonstrated elevated mRNA levels of IL-1α (*IL1A*) in MDR-*Kpneu*, meropenem, and flagellin-exposed HBE cells (Appendix A), protein levels of IL-1α in basolateral supernatant or apical wash were below the limit of detection. Taken together, these findings indicate that flagellin augments the secretion of chemokines during MDR-*Kpneu* infection but only in the absence of meropenem treatment.

### 2.3. Secretion of Inflammatory Mediators by HBE Cells Activates Polymorphonuclear Cells and Monocytes

Host defense against bacteria in the airways is mediated by a coordinated action of the respiratory epithelium and leukocytes recruited to the site of infection from the circulation [2]. In view of the profound release of chemokines by HBE cells after infection with MDR-*Kpneu* and treatment with meropenem and flagellin, we next assessed whether HBE supernatant contained factors capable of activating blood leukocytes. Therefore, we incubated whole blood from healthy donors with HBE cell supernatants for 6 h and analyzed polymorphonuclear cell (PMN) and monocyte activation using CD11b surface expression as readout (Figure 3A,B). Supernatants from HBE cells exposed to MDR-*Kpneu* alone did not induce activation of PMNs or CD14^+^ monocytes (Figure 3C). Supernatants from HBE cells infected with MDR-*Kpneu* and treated with flagellin, however, increased CD11b expression on PMNs and CD14^+^ monocytes. Supernatants from HBE cells infected with MDR-*Kpneu* and treated with meropenem also triggered PMN activation, but this was not further enhanced by flagellin. In contrast to PMN activation, supernatants from HBE cells infected with MDR-*Kpneu* and treated with meropenem did not enhance CD11b expression on CD14^+^ monocytes as compared to supernatants harvested after MDR-*Kpneu* infection alone, but activation of these cells was enhanced by HBE supernatants obtained after flagellin treatment. Analysis of CD16^+^ monocytes revealed that none of the supernatants of stimulated HBE cells enhanced CD11b expression on this monocyte subset. To elucidate that the stimulatory effect of supernatants from flagellin-treated HBE cells was not triggered directly by flagellin, we examined the effect of flagellin in our whole-blood activation model. These experiments revealed no direct effect of flagellin on PMN activation and a small effect on CD14^+^ monocyte activation (Appendix A). Furthermore, since LPS is a direct stimulus of PMN and CD14^+^ monocyte activation (Appendix A), we also assessed whether LPS release by *Klebsiella* in the supernatant of MDR-*Kpneu*-infected HBE cells triggered the expression of CD11b on PMNs and CD14^+^ monocytes using the LPS inhibitor polymyxin B [22]. Whereas the addition of polymyxin B completely inhibited PMN and CD14^+^ monocyte activation by LPS, the inhibitor did not alter enhanced CD11b expression on PMNs and CD14^+^ monocytes by supernatants of MDR-*Kpneu*-infected and meropenem- and flagellin-treated HBE cells (Figure 3D). Taken together, these findings indicate that flagellin treatment of MDR-*Kpneu*-infected HBE cells augments the secretion of inflammatory mediators that are able to activate blood PMNs and CD14^+^ monocytes.

## 3. Discussion

Respiratory infections caused by *K. pneumoniae* form a major health risk, specifically in hospitalized patients or in those with underlying diseases [2,23]. Moreover, carbapenem-resistant *K. pneumoniae* contributes to deaths attributable to antimicrobial resistance worldwide [4]. To determine whether flagellin has the capacity to serve as immunostimulatory adjuvant treatment during antibiotic-treated infection in humans, we infected HBE cells with MDR-*Kpneu* and treated the cells with meropenem, flagellin, or a combination. Flagellin induced the production of chemokines and upregulated transcription of antibacterial defense genes during MDR-*Kpneu* infection. In combination with meropenem, however, flagellin did not augment these responses. Furthermore, we found that inflammatory mediators, secreted by flagellin-stimulated HBE cells, increased the activation of PMN and CD14^+^ monocytes. These results suggest that flagellin therapy may have potential to boost immune responses in humans during infection of the airways.

Previously, our group showed enhanced production of inflammatory mediators by HBE cells upon stimulation with flagellin or *P. aeruginosa,* a flagellated pathogen [12]. In line with these findings, we here demonstrate higher mRNA expression and secretion of inflammatory mediators by HBE cells stimulated with flagellin during MDR-*Kpneu* infection or flagellin alone. These results corroborate earlier studies, describing flagellin to evoke an inflammatory response via TLR5 signaling and the recruitment of phagocytes, such as neutrophils [10]. Moreover, during mouse pneumococcal pneumonia, combined treatment with flagellin and antibiotics augmented the release of Cxcl1 and Ccl20 in bronchoalveolar lavage fluid, the influx of neutrophils, and host defense, as reflected by a reduced bacterial burden in the lung [17]. To assess whether flagellin also triggers the secretion of inflammatory factors capable of activating human leukocytes, we incubated HBE cell supernatants with whole blood and analyzed surface CD11b expression. In line with enhanced secretion of neutrophil-attracting chemokines CXCL1 and CCL20, these experiments revealed that supernatant of MDR-*Kpneu*-infected HBE cells treated with flagellin activated human neutrophils and CD14^+^ monocytes. Further studies with intrapulmonary instillation in human volunteers [24] are required to establish the immunostimulatory capacity of flagellin in vivo.

An unexpected finding of our study was that apical exposure of HBE cells to MDR-*Kpneu* alone induced no or only a modest expression of antibacterial and inflammatory genes, suggesting that HBE cells do not sense live MDR-*Kpneu* as a danger signal. The hypercapsule of K2 *Klebsiella* strains is a crucial virulence factor that impedes phagocytosis, antimicrobial proteins, and complement-mediated lysis [23]. Interestingly, an earlier study also found that live *Klebsiella* K2:O1 did not upregulate the expression of antimicrobial genes in primary human bronchial cells, whilst an isogenic mutant strain lacking the polysaccharide capsule did increase the expression of human β–defensin 2 (hBD2) and hBD3 [25]. These findings imply that the polysaccharide capsule of K2 *Klebsiella* strains evades recognition of bacterial ligands by PRRs. In contrast to our findings with MDR-*Kpneu* alone, treatment of MDR-*Kpneu*-infected HBE cells with meropenem on the basolateral side triggered robust antimicrobial and inflammatory responses. It is known that β-lactam antibiotics, such as meropenem, can damage the bacterial cell wall of *Klebsiella* and cause the release of endotoxins even when concentrations are lower than the MIC [26]. It is, however, highly unlikely that endotoxin release from *Klebsiella* is the cause of epithelial activation in our experiments, as we and others previously showed no induction of an inflammatory response in primary lung epithelial cells after stimulation with LPS [12,13]. Of note, in an earlier study, we demonstrated that *K. pneumoniae* inactivated by UV irradiation (which only damages bacterial DNA and RNA strands) did increase the expression and secretion of chemokines by HBE cells [12]; this, taken together with the strong effect of (basolateral) meropenem on chemokine production by infected HBE cells in the absence of a measurable effect on apical bacterial counts, suggests that minor damage to *Klebsiella* is sufficient to activate HBE cells. The TLR2 ligand peptidoglycan-associated lipoprotein (PAL) [27], which is an outer-membrane protein of hypervirulent *Klebsiella* [28], may be considered in this respect since TLR2 is expressed by primary human lung epithelial cells [13,29]. Further studies are needed, however, to determine whether PAL or other bacterial ligands exert the immunostimulatory effect of MDR-*Kpneu* in the presence of antibiotics and the role of various TLRs in our model.

Besides chemokines, lung epithelial cells produce peptides, like β-defensins and LL-37, that exert direct microbicidal effects and also play a role in the recruitment of circulating immune cells [9,30,31]. In the current study, we found that flagellin alone, as well as flagellin in combination with meropenem, induced the expression of *DEFB4A*, encoding hBD2, in MDR-*Kpneu*-infected HBE cells. Earlier research revealed that hBD2 markedly decreased *Klebsiella* K2:O1 counts in vitro [25]. In our experiments, however, the increase in *DEFB4A* expression was not matched with a decrease in bacterial outgrowth. These seemingly contradictory results can be explained by differences in hBD2 concentrations in the supernatants of HBE cells and the planktonic killing assay, as well as other experimental characteristics of these studies. Furthermore, we found that flagellin upregulated the expression of *S100A8* and *S100A9* in HBE cells and the secretion of calprotectin on the apical side. In addition, during MDR-*Kpneu* infection, flagellin and meropenem treatment also increased the expression of *S100A8* and *S100A9*, but this was not accompanied by enhanced secretion of calprotectin. In patients with inflammatory lung disease or pneumonia, high levels of calprotectin are found in the lung, which contribute to host defense or homeostasis [21,32]. The antibacterial effect of calprotectin has been reported to rely on the reduction in metal availability, including zinc and iron, for various bacteria [21,33]. Our group showed that calprotectin reduced the growth of *K. pneumoniae* in vitro by metal chelation and that the addition of zinc to calprotectin reversed this effect [20]. The fact that, in the current study, we did not find a reduction in bacterial counts following the increase in calprotectin concentrations may be explained by the low concentration of calprotectin in the apical wash of HBE cells, which was almost a thousand times lower as compared to the concentration required to reduce the outgrowth of *Klebsiella* in vitro [20].

Our study has several limitations. First, we conducted our experiments with one *Klebsiella* strain. This capsular K2 strain is, like K1 strains, hypermucoid and capable of evading multiple host defense mechanisms [23]. Further studies with different capsular serotypes of *Klebsiella*, and other Gram-negative and Gram-positive bacterial strains, are required to demonstrate whether our conclusions also apply to other bacteria. Second, we used only one dosage of bacteria, flagellin, and meropenem in our studies. Massive outgrowth of MDR-*Kpneu* may account for the robust stimulation of HBE cells treated with meropenem alone. Further experiments with appropriate doses of MDR-*Kpneu* and flagellin may reveal whether flagellin in combination with meropenem has potential to boost the innate immune response of HBE cells. Third, our HBE model, while mimicking the mixed cellular composition of the human airways, lacks the presence of innate immune cells. PMNs play an important role in host defense against *K. pneumoniae* [34], by the release of high amounts of calprotectin and other antimicrobial proteins upon activation and the induction of neutrophil extracellular trap formation, aiding in limiting *Klebsiella* growth [20,21,35,36]. Future studies using organoids or advanced ex vivo models, such as precision-cut lung slices [37], supplemented with innate immune cells, could shed light on the interplay between the human lung epithelium and blood leukocytes in antibacterial defense against *K. pneumoniae*.

## 4. Materials and Methods

### 4.1. HBE Cell Culture

Primary human bronchial epithelial (HBE) cells were obtained anonymously from patients undergoing a lobectomy for lung cancer at the Amsterdam University Medical Centers (AUMC), the Netherlands. Healthy lung tissue was isolated distant from tumorous tissue by a pathologist, and the absence of malignant cells was confirmed by microscopy. The study protocol (2015-122#A2301550) was approved by the Institutional Review Board of AUMC, and written informed consent was obtained from the patients before sampling. Primary epithelial cells were isolated and differentiated following Fulcher’s protocol as described [8]. Briefly, passage 2 to 4 primary HBE cells were expanded in human type IV placental collagen-coated porous support 24-well Transwell inserts (Corning, Amsterdam, The Netherlands) for differentiation in PneumaCult-Ex Plus media (StemCell Technologies, Vancouver, BC, Canada) at 37 °C, 5% CO_2_. Once confluent, the medium on the basolateral side was changed to PneumaCult-ALI medium (further referred to as ALI medium; StemCell Technologies) containing penicillin and streptomycin, and the medium on the apical side was removed, thereby forming an ALI. Basolateral media were renewed every 2–3 days until full differentiation. Seven days before infection, media were depleted of antibiotics to avoid interference with the infection. These pseudostratified HBE cell layers express various TLRs that recognize bacterial ligands (Appendix A).

### 4.2. Treatment of HBE Cells

For infection of HBE cells, an isogenic carbapenem-resistant variant of *K. pneumoniae* American Type Culture Collection 43816 (K2:O1) was used, KPC EMC2014 strain (referred to in this study as MDR-*Kpneu*), generated via bacterial conjugation as described [38]. MDR-*Kpneu* was grown in Tryptic Soy Broth medium to log-phase as described [39] and extensively washed with PBS prior to infection. MDR-*Kpneu* was capable of growing in ALI medium and susceptible to 50 μg/mL meropenem (Fresenius Kabi, Huis ter Heide, The Netherlands), >6 fold the MIC of 8 μg/mL (Appendix A). HBE cells were infected on the apical surface with 1000 CFUs of MDR-*Kpneu* in 20 μL volume of PBS or treated with PBS as control. Six hours after bacterial infection, cells were treated with either 50 μg/mL meropenem in the basolateral medium or medium alone. Simultaneously, cells were treated on the apical side with 0.1 μg flagellin in 20 μL PBS or vehicle (PBS). Recombinant flagellin (Statens Serum Institut, Copenhagen, Denmark), originating from *Salmonella enterica* serovar Typhimurium (ATCC14028), was produced as described [40]. After 24 h of bacterial infection, the apical side of the HBE cell layer was washed with 200 μL PBS, the basolateral medium was collected, and cells were lysed in Lysis/Binding Buffer (Roche, Basel, Switserland) and stored at −80 °C for RNA isolation. Apical wash and basolateral media were serially diluted 10-fold, and 50 μL was plated on blood agar plates in order to determine bacterial loads. Apical wash and culture supernatants were stored at −20 °C until analysis.

### 4.3. ELISA

CXCL1, CXCL8, CCL20, and calprotectin in basolateral medium and calprotectin in apical wash were measured by ELISA, according to the manufacturer’s instructions (R&D Systems, Minneapolis, MN, USA).

### 4.4. mRNA Analysis

For analysis of mRNA levels of antimicrobial proteins and chemokines, total mRNA was isolated from HBE cells using the High Pure RNA Isolation Kit (Roche, Basel, Switzerland), following the manufacturer’s instructions. cDNA synthesis and qPCR were performed as previously described [12]. Data were analyzed with LinRegPCR (11.0) software based on PCR efficiency values derived from amplification curves. HPRT1 mRNA levels were used for normalization. All primers are listed in Table 1.

For analysis of mRNA levels of TLRs in HBE cells, publicly available RNA sequencing data, i.e., GSE164704, from unstimulated HBE cells of two different donors were analyzed as described [12].

### 4.5. Peripheral Blood Leukocyte Activation

Supernatants of HBE cells were assessed for the presence of leukocyte activation factors using a whole-blood stimulation assay. Heparinized blood was obtained from healthy volunteers (Amsterdam UMC medical ethical committee approval No. 2015_074). Basolateral HBE supernatant of technical replicates was pooled, and aliquots were added to a similar volume of freshly drawn blood in polypropylene plates and incubated at 37 °C, 5% CO_2_. Whole-blood incubation with plain ALI culture medium, or supplemented with LPS 10 ng/mL, LPS and polymyxin 50 μg/mL (invivogen, San Diego, CA, USA), or flagellin 1 μg/mL, was used as a control. After 6 h of stimulation, leukocytes were stained according to the manufacturer’s recommendations with anti-human CD66b PerCP/Cyanine5.5 (clone G10F5), anti-human CD11b APC (clone ICRF44), anti-human CD14 FITC (clone M5E2), and anti-human CD16 PE-Cy7 (clone 3G8, all BD biosciences, Franklin Lakes, NJ, USA), after which the cells were fixed, and erythrocytes were lysed using Fix/Lyse solution (ThermoFisher, Waltham, MA, USA). Leukocyte activation was determined by flow cytometry using a Cytoflex-S (Beckman Coulter, Brea, CA, USA), and data were analyzed using FlowJo v10 software (Becton Dickinson, Franklin Lakes, NJ, USA). The gating strategy is shown in Appendix A. PMNs were defined as CD66b^+^, and monocytes were defined as either CD14^+^ or CD16^+^. Cell activation of PMNs and CD14^+^ monocytes was determined by the mean fluorescence intensity (MFI) of CD11b on both cell types.

### 4.6. Statistical Analysis

Statistical analysis was performed using a mixed model with a random intercept on experiment level with 3–4 technical replicates from 3 separate experiments. All (co)variances were assumed equal to each other and equivalent to a compound symmetry covariance structure. In case the overall group difference was significant, determined by type-III Wald test in the mixed model analyses (*p* < 0.05), then each stimulation was compared with the reference category to identify particular differences between groups. Data from whole-blood stimulation experiments with multiple healthy donors were analyzed using a paired *t*-test in GraphPad Prism version 9 (Graphpad Software, San Diego, CA, USA). Statistical significance is shown as */# *p* < 0.05, **/## *p* < 0.01, ***/### *p* < 0.001, ****/#### *p* < 0.0001.

## 5. Conclusions

Flagellin treatment of MDR-*Kpneu*-infected HBE cells augments the expression of antimicrobial factors and chemokines and induces the secretion of mediators that activate PMNs and monocytes. These findings suggest that adjunctive flagellin treatment may have potential to boost innate immune responses in the lung during respiratory infection.

## Figures and Tables

**Figure 1 ijms-25-00309-f001:**
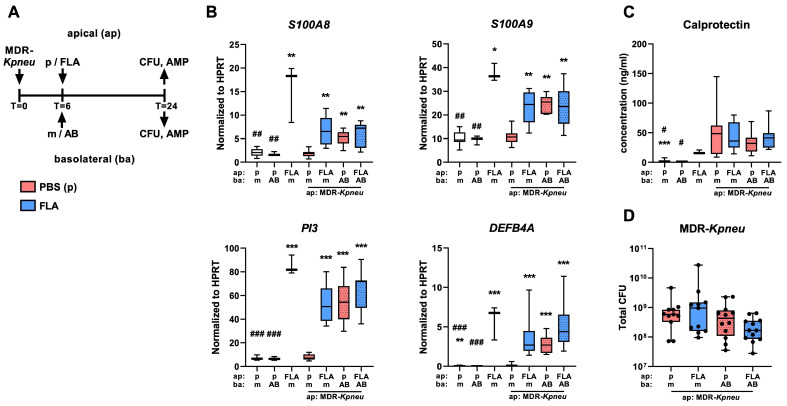
Flagellin augments the expression of antimicrobial proteins by MDR-*Kpneu*-infected HBE cells but does not affect bacterial outgrowth. Human bronchial epithelial (HBE) cells were infected on the apical (ap) side with 1000 colony-forming units (CFUs) of multi-drug-resistant (MDR) *Klebsiella pneumoniae* (*Kpneu*) and treated 6 h later with 50 μg/mL meropenem (AB) or medium (m) in the basolateral (ba) medium and 0.1 μg flagellin (FLA; blue) or phosphate-buffered saline (PBS; red) (p) on the apical side; samples were collected 24 h after infection (**A**). mRNA levels of β-defensin 4a (*DEFB4A*), peptidase inhibitor 3 (*PI3*), and calprotectin (*S100A8* and *S100A9*) relative to the housekeeping gene *HPRT* in HBE cells (**B**). Protein levels of calprotectin (S100A8/A9 heterodimer) in the apical wash (**C**). CFU counts of MDR-*Kpneu* in the apical wash (**D**). Data are shown as box and whiskers representing 3 repeated experiments with n = 4 per group. # indicates significant differences as compared to FLA alone; * indicates significant differences as compared to MDR-*Kpneu* + medium treated cells. */# *p* < 0.05, **/## *p* < 0.01, ***/### *p* < 0.001.

**Figure 2 ijms-25-00309-f002:**
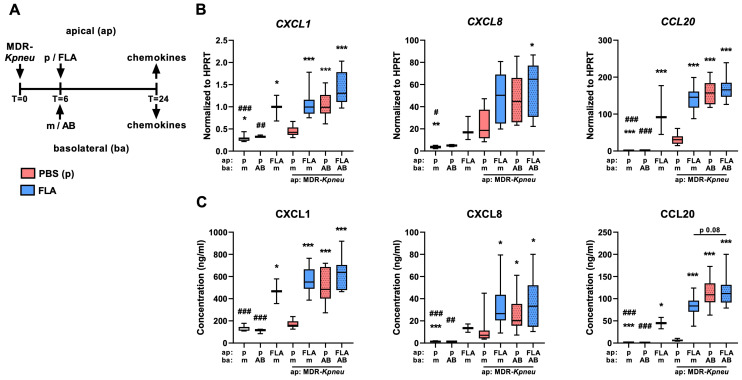
Flagellin augments the secretion of chemokines during MDR-Kpneu infection in the absence of meropenem treatment. HBE cells were infected with MDR-Kpneu on the apical (ap) side and treated at t = 6 h with meropenem (AB) or medium (m) in the basolateral (ba) medium and flagellin (FLA; blue) or PBS (p; red) on the apical side; samples were collected 24 h after infection (**A**). Expression levels of CXCL1, CXCL8, and CCL20 normalized to HPRT in HBE cells (**B**). Protein levels of CXCL1, CXCL8, and CCL20 in basolateral (ba) medium (**C**). Data are shown as box and whiskers representing 3 repeated experiments with n = 4 per group. # indicates significant differences as compared to FLA alone; * indicates significant differences as compared to MDR-Kpneu + medium treated cells. */# *p* < 0.05, **/## *p* < 0.01, ***/### *p* < 0.001.

**Figure 3 ijms-25-00309-f003:**
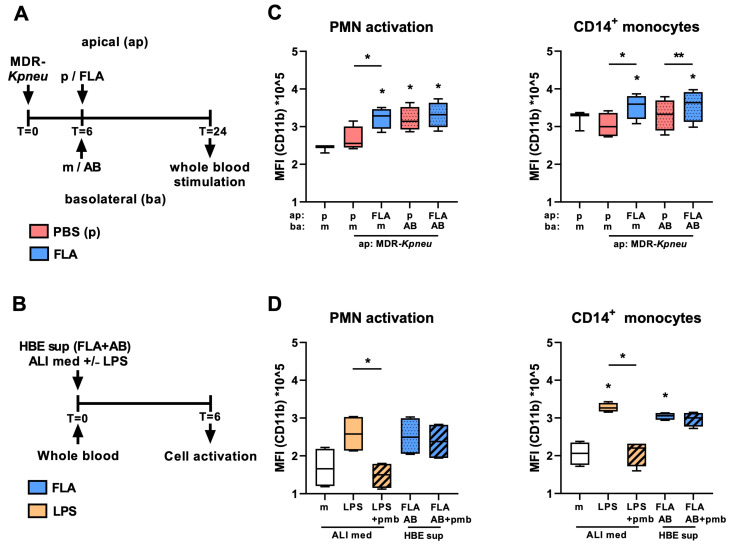
Flagellin treatment of MDR-*Kpneu*-infected HBE cells augments the secretion of inflammatory mediators that activate PMNs and CD14^+^ monocytes in whole blood. HBE cells were infected with MDR-*Kpneu* on the apical (ap) side and treated at t = 6 h with meropenem (AB) or medium (m) basolaterally and flagellin (FLA; blue) or PBS (p; red) apically; basolateral supernatants were collected 24 h after infection (**A**). Supernatants of HBE cells or ALI medium, supplemented with 10 ng/mL lipopolysaccharide (LPS; orange) or not, were incubated with whole blood for 6 h after which PMN (CD66b^+^) and CD14^+^ monocyte activation (mean fluorescence intensity (MFI) of CD11b) was determined by flow cytometry (**B**). Gating strategy is shown in Appendix A. Activation of PMNs and CD14^+^ monocytes by supernatant of MDR-*Kpneu*-infected HBE cells after different treatments as described in A (**C**). Activation of PMNs and CD14^+^ monocytes by ALI medium supplemented with LPS with and without 5 μg/mL polymyxin B (pmb) or supernatant of MDR-*Kpneu*-infected HBE cells treated with meropenem and flagellin with and without polymyxin B (**D**). Data are shown as box and whiskers representing data from 2–4 healthy blood donors with 4 technical replicates per group derived from pooled supernatants. * Indicates significant differences as compared to medium-treated controls or between the indicated groups. * *p* < 0.05, ** *p* < 0.01.

**Table 1 ijms-25-00309-t001:** Primer sequences used for qPCR.

Gene	Forward	Reverse
*HPRT*	GGATTTGAAATTCCAGACAAGTTT	GCGATGTCAATAGGACTCCAG
*CXCL1*	GCATACTGCCTTGTTTAATGGT	CCAGTAAAGGTAGCCCTTGTTTC
*CXCL8*	AACCTTTCCACCCCAAATTTAT	AAAACTTCTCCACAACCCTCTG
*CCL20*	TGCTGTACCAAGAGTTTGCTC	CGCACACAGACAACTTTTTCTTT
*DEFB4A*	CTCCTCTTCTCGTTCCTCTTCA	GCAGGTAACAGGATCGCCTAT
*S100A8*	GGGAATTTCCATGCCGTCTA	GACGTCTGCACCCTTTTTCC
*S100A9*	GAACCAGGGGGAATTCAAAG	CCAGGTCCTCCATGATGTGT
*PI3*	TTGATCGTGGTGGTGTTCCT	GAACACGGCCTTTGACAGTG

## Data Availability

The data presented in this study are available upon request from the corresponding author.

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
