# Peer review of "Immunostimulatory Effect of Flagellin on MDR-Klebsiella-Infected Human Airway Epithelial Cells"

_ijms, 2023, doi:10.3390/ijms25010309_

Round 1
Reviewer 1 Report
Comments and Suggestions for Authors
The authors examined the effect of flagellin on MDR-Kpneu infected primary human bronchial epithelium cell culture. The authors found potential use of flagellin in MDR-Kpneu infection. The idea is interesting and important. Comments for the authors below:
Major points:
1. Line 65: Please include reference for key PRR on the airway epithelium and explain why the authors focused only TLR5 but nor other TLRs.
2. Can the authors try to knockdown TLR5 in HBE cells with or without antibiotic? The role of TLR5 in this model system is not clear.
3. Is the expression level of TLR5 in HBE cells known?
Minor points:
1. Line 240: Chemokines should be described in capital for consistency.
2. Line 267: CO2, 2 should be subscript.
Author Response
We thank the reviewer for the positive remarks on our study and manuscript.
- Please include reference for key PRR on the airway epithelium and explain why the authors focused only TLR5 but nor other TLRs (line 65).
Reply: We thank the reviewer for this remark. As we pointed out in the Introduction (line 76-77), new treatment strategies are being developed to aid in host defense against antibiotic-resistant bacteria. Our research is performed as part of the research of the FAIR consortium (https://fair-flagellin.eu/) which aims to investigate the beneficial effect of additional and topical (pulmonary) administration of flagellin during pneumonia. Since flagellin is known to trigger TLR5 activation in the lung and in lung epithelial cells, we have included the literature on this topic in the Introduction (line 64-75). The focus of our study, however, is not on TLR5 but on the immunostimulatory effect of flagellin on human bronchial epithelial cells. Previously, we have shown (PMID: 33542495) that human bronchial epithelial cells are not activated by the TLR4 ligand LPS. TLR2, however, is expressed by human bronchial epithelial cells and this was described in the Discussion (line 286-289) in relation to our unexpected finding that meropenem treatment of MDR-Kpneu infected bronchial epithelial cells resulted in cell activation.
To address this comment by the reviewer, we now analyzed mRNA levels of various TLRs involved in bacterial recognition in the current human bronchial epithelial cell model using RNAseq data which we published previously (PMID: 33542495). This analysis revealed that these pseudo-stratified human bronchial epithelial cells show clear expression of TLR1, TLR2, TLR3, and TLR5, and to a lesser extent TLR4 and TLR6, while mRNA levels of TLR7 and TLR8 in HBE cells were at the limit of detection.
Figure S5. Expression of different Toll-like-receptors (TLRs) in HBE cells. Total mRNA counts of TLRs as determined by RNA sequencing analysis described in reference 12 (accession number for the RNA sequencing data is GEO: GSE164704). Data represent duplicates of unstimulated HBE cells from two different donors.
We have included these data in the supplementary material (Figure S5) and adjusted the text (line 427-428). This analysis was mentioned in the Methods (line 381-383) Furthermore, we have added the following sentence to the manuscript (line 347-348).
“These pseudostratified HBE cell layers express various TLRs that recognize bacterial ligands (Figure S5).”
In addition, we have adjusted the text about TLRs (line 64-67) and added an additional reference for this topic (Leiva-Juárez et al., PMID: 28812547).
- Can the authors try to knockdown TLR5 in HBE cells with or without antibiotic? The role of TLR5 in this model system is not clear.
Reply: Although we agree with the reviewer that it would be interesting to further study the role of TLR5 in our human bronchial epithelial cell model, we consider this question beyond the scope of our study, which focuses on the stimulatory effect of flagellin in an infectious environment and not on the role of particular TLRs.
To address this comment by the reviewer, however, we have added the following sentence to the Discussion (line 291)
“Further studies are needed, however, to determine ......... and the role of various TLRs in our model.”
- Is the expression level of TLR5 in HBE cells known?
Reply: As described above (comment 1) we have now included the analysis of TLR mRNA levels in our human bronchial epithelial cell model to the results (Figure S5) and adjusted the text. Indeed TLR5 is expressed in our model.
- Chemokines should be described in capital for consistency (line 240)
Reply: The guidelines for writing proteins from different species is that human proteins are written in upper-case (e.g. CXCL1) and that for mouse proteins only the first letter is written in upper-case (e.g. Cxcl1). Since the sentence in line 240 describes the results from a mouse study, these proteins are written with only a capital first letter.
For guidelines, see for example:
www.biosciencewriters.com/Guidelines-for-Formatting-Gene-and-Protein-Names.aspx
- CO2, 2 should be subscript (line 267)
Reply: We thank the reviewer for pointing us to this mistake and we have corrected the text (line 380).

Reviewer 2 Report
Comments and Suggestions for Authors
This very complex study deals with the role of flagellin in lung infections and is proposed as a possible improvement to treat antibiotic-resistant bacteria. Tha paper is very confusing and difficult to read and understand also as regards the figures. Authors have to explain the reasons they decide to conduct some experiments that are presented in a non-clear global study. As an example the role of flagellin together with meropenem is not clear. what is the role of meropenem in stimulating the production of the citokines? Authors stress that flagellin is a potential immunostimulatory adjuvant but how is functioning together with the antibiotic? All these aspects have to be cleared. In general, the text has to be simplified and make it more easily to understand
line 99: please correct the sentence
Figures: the meaning of the colours is not explained in the legend
The legend to Figure 2 is very confusing, please try to get it more clear
In general in the figures the indications in the abscissa axis are not clear
Comments on the Quality of English Language
English quality seems to be OK
Author Response
We thank the reviewer for the remarks on our study and regret that the reviewer finds our manuscript very confusing and difficult to read and understand.
- Authors have to explain the reasons they decide to conduct some experiments that are presented in a non-clear global study. As an example the role of flagellin together with meropenem is not clear.
Reply: Although we do not exactly understand this point of the reviewer, we have adjusted the text of the Introduction to clarify the use of flagellin in combination with meropenem.
The rationale for using a combination of meropenem and flagellin is that patients with bacterial pneumonia will always receive antibiotics from a doctor as a first choice of treatment. The aim of our study is to determine whether additional, local (pulmonary) treatment with flagellin improves the host response against bacterial infection. Therefore, to mimic the clinical situation, we used a model with MDR-Kpneu infection on the apical side (representing the airways) and then applied antibiotic treatment during ongoing infection on the basolateral side (representing the inside of the body) and administered flagellin on the apical side in view of (topical) aerosol therapy.
To address this comment, we have added the following sentence and reference regarding our choice of treatment with meropenem (PMID 18416587) to the Introduction (line 84-90).
“. Since antibiotics are the mainstay of therapy for bacterial pneumonia [2], we investigated the effect of flagellin also in combination with meropenem, a broad-spectrum b-lactam antibiotic, often used as empirical therapy [19]. To mimic the clinical situation, we commenced treatment with flagellin and/or meropenem during ongoing infection with MDR-Kpneu on the apical side of the HBE cells (representing the airways) and applied meropenem to the basolateral medium (representing the inside of the body) and flagellin on the apical side in view of topical therapy.”
- What is the role of meropenem in stimulating the production of cytokines?
Reply: Like the reviewer, we do not know why meropenem treatment triggers the secretion of cytokines by infected human bronchial epithelial cells. In the Discussion (line 275-289), we have provided possible explanations for this finding.
As we have already indicated (line 289-291), further investigation is required to determine which stimulatory components of MDR-Kpneu are released as a result of treatment with meropenem. Unfortunately, however, our laboratory is not experienced and equipped for experiments to identify such bacterial ligands. Therefore, we have chosen to not further pursue this issue.
- Authors stress that flagellin is a potential immunostimulatory adjuvant but how is functioning together with the antibiotic?
Reply: We agree with the reviewer that our results do not demonstrate that flagellin is a potential immunostimulatory adjuvant in combination with meropenem. The observations that (I) treatment of MDR-Kpneu infected HBE cells with meropenem alone induces a robust inflammatory response (which is not further augmented by flagellin), and that (II) MDR-Kpneu demonstrates a massive outgrowth on HBE cells (even in the presence of meropenem), suggest that the bacterial dosage used in the current experiments is inappropriate to demonstrate an additional effect of flagellin under these conditions. We anticipated that experiments with lower doses of MDR-Kpneu may reveal whether flagellin has potential as an immunostimulatory adjuvant in combination with antibiotics. However, concerning the time required for culturing pseudo-stratified human bronchial epithelial cells, it is not possible to perform these experiments within the time given us for the resubmission.
- In general, the text has to be simplified and make it more easily to understand.
Reply: We are sorry that the reviewer finds our manuscript difficult to understand, but we are unsure how to address this comment. We have adjusted the text of the Introduction (see comment 1) to better explain the rationale and design of our study and we have adjusted Figures 2 and 3 with a graphical scheme of the experiments, as well as the timing and side of administration of reagents, to make them easier to understand (also see comment 7, below). We hope that these changes make the manuscript more easy to understand.
- Please correct the sentence (line 99)
Reply: We have adjusted this sentence.
- Figures: the meaning of the colours is not explained in the legend
Reply: We have adjusted Figure 2 and 3 to explain the meaning of the colors.
- The legend to Figure 2 is very confusing, please try to get it more clear
Reply: We have added a graphical scheme of the experiments in both Figure 2 and Figure 3 to better explain the treatment of the HBE cells regarding the timing and side of administration of reagents and have adjusted the legends of both figures.
- In general in the figures the indications in the abscissa axis are not clear
Reply: As described above (comment 7) we have added a graphical scheme of the experiments in both Figure 2 and 3 to better explain the indications on the X-axis.

Round 2
Reviewer 2 Report
Comments and Suggestions for Authors
Responses to my review are OK, however the way flagellin and meropenem are working together is not still explained in the text. Would be interesting to explain this point but I dont need to review again the paper
Author Response
Reply to Reviewer 2:
(Please note that line numbers indicated in this reply refer to line numbers in the revised version of the manuscript with marked changes)
We thank the reviewer for the remark on our study.
- The way flagellin and meropenem are working together is not still explained in the text. Would be interesting to explain this point.
Reply:
We are unsure how to address this comment by the reviewer, considering our results do not show an augmented immune response when flagellin treatment is combined with meropenem on infected HBE cells. Since these results do not point that flagellin and meropenem are working together, it is difficult to provide an explanation for something we did not observe.
In the Introduction (line 76-84), we have explained the rationale to use flagellin in combination with meropenem in our model and the beneficial effect of flagellin in combination with antibiotics in pneumococcal pneumonia in mice.
To further explain the working of flagellin and antibiotics together we have elaborated on the effect of flagellin in the latter study and added the following sentence (line 80-83):
“These studies showed that flagellin in combination with antibiotics augmented the chemokine response in the lungs as compared to treatment with antibiotics alone, which was associated with increased neutrophil recruitment to the lung [17, 18].“
Furthermore, to clarify in the Discussion that we did not find that flagellin and meropenem are working together in the current study, we have added the following sentence (line 244-245):
“In combination with meropenem, however, flagellin did not augment these responses.”
Moreover, in the Discussion (line 268-295) we have explained the unexpected finding that exposure of HBE cells to MDR-Kpneu alone induced no or only a modest inflammatory response, whereas meropenem treatment of infected HBE cells evoked a robust response.
In our first rebuttal, we have explained that the bacterial dosage used in our experiments may be inappropriate (too high) to demonstrate an additional effect of flagellin under these conditions, and that we anticipated that experiments with lower doses of MDR-Kpneu may reveal whether flagellin has potential as an immunostimulatory adjuvant in combination with antibiotics. We have added this argumentation as a limitation of our study to explain the lack of effect of flagellin in combination with meropenem. We have added the following sentence (line 325-330):
“Second, we used only one dosage of bacteria, flagellin and meropenem in our studies. Massive outgrowth of MDR-Kpneu may account for the robust stimulation of HBE cells treated with meropenem alone. Further experiments with appropriate doses of MDR-Kpneu and flagellin may reveal whether flagellin in combination with meropenem has potential to boost innate immune response of HBE cells. Third, ……”
